# Frailty Assessment in a Cohort of Elderly Patients with Severe Symptomatic Aortic Stenosis: Insights from the FRailty Evaluation in Severe Aortic Stenosis (FRESAS) Registry

**DOI:** 10.3390/jcm10112345

**Published:** 2021-05-27

**Authors:** Pablo Solla-Suárez, Pablo Avanzas, Isaac Pascual, Manuel Bermúdez-Menéndez De La Granda, Marcel Almendarez, Jose M. Arche-Coto, Daniel Hernández-Vaquero, Rebeca Lorca, Eva López-Álvarez, Rut Álvarez-Velasco, Carmen Moreno-Planillo, César Morís de la Tassa, José Gutiérrez-Rodríguez

**Affiliations:** 1Área de Gestión Clínica de Geriatría, Hospital Monte Naranco, 33011 Oviedo, Spain; pasosu@gmail.com (P.S.-S.); manuelbermudezmg@hotmail.com (M.B.-M.D.L.G.); jmarche74@hotmail.com (J.M.A.-C.); evam.lopez@sespa.es (E.L.-Á.); mcarmen.moreno@sespa.es (C.M.-P.); 2Instituto de Investigación Sanitaria del Principado de Asturias, 33011 Oviedo, Spain; avanzas@gmail.com (P.A.); marcel.almendarez@gmail.com (M.A.); dhvaquero@gmail.com (D.H.-V.); lorcarebeca@gmail.com (R.L.); rutalvarez3@gmail.com (R.Á.-V.); cesarmoris@gmail.com (C.M.d.l.T.); pepe.gutierrez@telefonica.net (J.G.-R.); 3Área de Gestión Clínica del Corazón, Hospital Universitario Central de Asturias, 33011 Oviedo, Spain; 4Departamento de Medicina, Universidad de Oviedo, 33011 Oviedo, Spain; 5Departamento de Biología Funcional, Universidad de Oviedo, 33011 Oviedo, Spain

**Keywords:** transcatheter aortic valve replacement (TAVI), Transcatheter Aortic Valve Implantation (TAVR), frailty, aortic stenosis

## Abstract

Background: Precise evaluation of the degree of frailty is a fundamental part of the global geriatric assessment that helps to avoid therapies that could be futile. Our main objective was to determine the prevalence of frailty in a specific consult of patients undergoing aortic valve replacement. Methods: From May 2018 to February 2020, all consecutive patients ≥75 years old, with severe symptomatic aortic stenosis, undergoing valve replacement in the Principality of Asturias (Northern Spain) were evaluated. Results: A total of 286 patients were assessed. The mean age was 84 ± 4.01 years old; 175 (61.2%) were female. The short performance physical battery score was 8.5 ± 2.4 and the prevalence of frailty was 19.6% (56 patients). In the multivariable analysis, age, Barthel index and atrial fibrillation were independent predictors of frailty. Conclusions: The prevalence of frailty in our sample patients undergoing aortic valve replacement, evaluated by a standardized protocol, was 19.6%.

## 1. Introduction

Frailty in elderly patients with severe aortic stenosis (AS) undergoing surgical or transcatheter aortic valve replacement (AVR) is an important prognostic factor. However, it has not been included in the usual risk score assessment scales to date [1,2]. Precise evaluation of the degree of frailty in AS patients is crucial as a fundamental part of the global geriatric assessment. It would help select the best intervention modality and identify cases in which an invasive treatment could be futile [3,4].

The evaluation of frailty could identify patients at risk of developing delirium, falls, or deconditioning during hospitalization. This evaluation is essential to select an adequate treatment and improve outcomes, such as length of stay, readmissions, functional deterioration and death in elderly patients with severe AS [5,6]. However, it is not always possible due to the different criteria, a lack of uniformity and the significant variability observed in previous studies regarding frailty assessment. Several scales, such as the “Short Performance Physical Battery” (SPPB), have correlated with the mortality risk and the functional deterioration of patients undergoing AVR. However, there is no consensus on the best way to assess frailty and even the risk factors to include in its definition remain controversial [7]. There are multiple proposals for frailty evaluation. In the present study, we assessed frailty with the widely recognized and internationally accepted SPPB scale. Previous studies regarding valvular intervention have shown that the SPPB scale can predict clinical outcomes after valve intervention [8,9,10,11,12,13,14,15]. 

The FRESAS study’s objective was to determine the actual prevalence in real clinical practice of frailty in patients ≥75 years old with severe symptomatic AS undergoing surgical or transcatheter AVR. We implemented a specific protocol, and all the characteristics of frailty were analyzed.

## 2. Materials and Methods

### 2.1. Study Population and Design

The (Frailty Evaluation in Severe Aortic Stenosis) FRESAS is an observational, longitudinal and prospective study. From May 2018–February 2020, patients ≥75 years with severe symptomatic AS undergoing AVR were included for analyses. The area of reference was the Principality of Asturias region (Northern Spain). A cut-off point of patients over 75 years was established based on the lower prevalence of frailty reported in younger subjects. From the 312 initial patients, 26 subjects were excluded from the study (9 patients rejected intervention, 9 patients underwent emergent aortic valve treatment before frailty evaluation and 8 patients died before frailty assessment).

All the included patients were diagnosed with severe symptomatic AS with indication for intervention according to the current clinical guidelines. They were all evaluated by a “heart team” to estimate the surgical risk and perform a global geriatric assessment. All participants signed the informed consent and the local ethics committee approved the study (Reference number 281/19). Patients <75 years or unable to sign the informed consent were excluded from the study.

According to the SPPB evaluation, we divided the patients into two groups: group A (frail, with an SPPB score of 6 or less) and group B (not frail, SPPB >6). 

### 2.2. Evaluation Protocol

All participants were discussed in a heart team meeting. Heart team participants included an interventional cardiologist, an expert in cardiac imaging, a cardiac surgeon, and a geriatrician using the geriatric evaluation information and all their baseline characteristics. We took advantage of the studied region’s geographical and sanitary organization (reference population of one million habitants and one center that provides invasive treatment for AS). Therefore, all the patients with severe symptomatic AS were evaluated by the same heart team, minimizing patient loss and granting a uniform evaluation. All patients ≥75 years that were candidates for AVR in our center were referred for specific frailty assessment. 

Our protocol included evaluating each subject by a nurse trained in geriatric evaluation with a dedicated scale to allow a prospective assessment of the different aspects that encompass frailty in an outpatient-based consult before any intervention modality was decided. The same person evaluated all the patients. Patients were given a phone number to notify of changes in their functional status.

The evaluating nurse acquired data from the clinical history and the other variables required to complete our protocol in the same visit. The following variables were recollected during the evaluation: -Demographics: age and sex-Clinical: most prevalent diseases, such as arterial hypertension (HT), diabetes mellitus (DM), dyslipidemia, atrial fibrillation (AF), acute coronary syndrome (ACS), heart failure (HF), chronic pulmonary obstructive disease (COPD), pulmonary hypertension (PH), peripheral vascular disease (PVD), previous cardiac surgery, stroke, mild cognitive impairment or dementia, thyroid pathology, chronic kidney disease (CKD), anemia and depression. Comorbidity was evaluated with the short-form Charlson index (Comorbidity was considered with a ≥2 score).-Cardiological: intervention risk assessment with EuroSCORE-II and echocardiographic data: left ventricle ejection fraction (LVEF), aortic valve area (AVA), mean and peak aortic gradient.-Blood workup: hemoglobin, albumin, creatinine, estimated glomerular filtration rate (eGFR), folic acid, thyroid-stimulating hormone (TSH) and vitamin D (25OH-D3)-Nutritional: “Mini Nutritional Assessment Short Form” (MNA) score.-Functional: functional status of the basic activities of daily living (BADL) were determined with the Barthel index that includes 10 items in its evaluation (Transfer, bowel, bladder, mobility, grooming, dressing, toilet use, stairs, feeding and bathing)and instrumental activities of daily living (IADL) with the Lawton index (differentiating maximum punctuation of 8 in women and 5 in men).-Cognitive and affective: “Minimental State Examination” (MMSE) and the “Geriatric Depression Scale of Yesavage” in its 15-item version.-Frailty assessment through the “Short Performance Physical Battery”.

### 2.3. SPPB

SPPB is one of the most precise scales to evaluate frailty with a substantial predictive capacity to identify a patient’s dependence degree, risk of institutionalization, rehospitalization, and mortality [16,17,18]. Furthermore, it has shown a significant correlation between the risk of mortality and functional deterioration in patients undergoing TAVR. [11,12,13,14,15]. This scale considers 3 items: equilibrium, walking speed and the ability to stand from a chair 5 times [19]. According to the SPPB scale, patients are classified as frail (0–6 points), prefrail [9,10,11] or robust (10–12 points) [20,21]. We divided the patients into two groups: group A (frail, SPPB 0–6) and group B (not frail, SPPB > 6). 

### 2.4. Statistical Analysis

Normality was assessed with the Kolmogorov–Smirnov (KS) test; the significance value for the KS normality test was established <0.05 for the studied variables. Quantitative continuous variables are compared using Student’s T-test or Mann–Whitney’s U-test and proportions with chi-square. Statistically significant differences were considered with a *p*-value <0.05. 

We performed a multivariable analysis to identify variables that could predict the study’s primary objective (presence of frailty) with linear regression, including variables associated with frailty with *p* <0.1. Statistical goodness was assessed with the Hosmer–Lemeshow test with *p* = 0.79 (significant *p* >0.05). Data were analyzed using SPSS version 25 (Chicago, IL, USA). Linear regression was performed considering SPPB as a continuous variable instead of dichotomic (frail and non-frail). The variables that predicted the SPPB score in the univariable analysis with *p* <0.1 using the backward stepwise method were included to perform a multivariable linear regression. 

## 3. Results

### 3.1. Baseline Characteristics

A total of 286 patients were evaluated and included for analysis. From the total patients, 72 (25.2%) were referred for surgical AVR and 179 (62.6%) for TAVR. In 35 (12.2%) patients, any intervention was deemed futile and conservative medical treatment was established. Baseline characteristics are shown in Table 1. The mean age was 84 ± 4 years old, from which 49.3% were >85 years old. There were 175 (61.2%) females. Regarding the cardiovascular risk factors, 221 (77.3%) patients had arterial hypertension, 156 (54.6%) presented dyslipidemia and 71 (24.8%) patients were suffering from type 2 diabetes mellitus. 

The mean AVA was 0.7 ± 0.2 cm^2^, the mean aortic gradient was 44.9 ± 14.4 mmHg and the peak gradient was 72.8 ± 22 mmHg. Furthermore, 170 (59.4%) patients presented with an adequate nutritional state, 191 (66.8%) patients had an average score in the MMSE and 207 (72.4%) patients did not present with a mood disorder evaluated with the Yesavage depression scale. Regarding the functional status, 76 (43.4%) patients among the women and 60 (54.1%) among the men were independent to execute the basic activities of daily living and personal care. 

The frailty assessment showed a mean score in the SPPB of 8.5 ± 2.4 and a prevalence of 19.6% (56 patients). Compared with frail patients, non-frail subjects had a higher prevalence of preserved LVEF. Moreover, in the blood workup, higher values of hemoglobin and albumin were observed. Frail patients presented with a worse nutritional, functional and mental situation, showing worse scores in the MNA, Lawton and Barthel index, MMSE and the Yesavage geriatric depression scale.

### 3.2. Multivariable Analysis

We performed a multivariable analysis with the variables that significantly predicted frailty with a logistic regression model and frailty as a categorical variable (frail/non-frail). In the final logistic regression model, age, AF and the Barthel index are significantly associated with frailty (Table 2). In the multivariable linear regression, being older (age), AF, a higher score in the Charlson index and the Yesavage depression scale, and a lower score in the Barthel index predict a lower SPPB (Table 3). 

## 4. Discussion

The main strength of the FRESAS study was the systematic evaluation of a homogenous and controlled sample of elderly patients with symptomatic severe AS referred for AVR. According to a standardized protocol, all participants were evaluated by the same trained person in a specific outpatient consult designed for this purpose. To our knowledge, this is the first study that systematically evaluates with a specific protocol the frailty in patients over 75 years old referred for aortic valve replacement. Therefore, we estimated the actual prevalence of frailty in these patients in routine clinical practice.

Current available surgical risk scales provide invaluable help in decision-making for each patient’s best intervention modality (i.e., surgical vs. TAVR). In our cohort of patients, 35 subjects did not receive any invasive treatment to avoid futility due to our frailty evaluation. The immediate and first-year outcomes are conditioned by other factors that the usual risk score assessment scales do not consider. Frailty is one of these factors, and it has been associated with the prognosis of patients undergoing surgical or transcatheter valvular intervention for the aortic and mitral valve [20,22]. 

Depending on the evaluating scale, the prevalence of frailty in the different registries varies [15]. Other physiological, nutritional, socio-economical and self-care variables may interfere with the evaluation and enhance this variability [7,8]. The SPPB is a widely recognized scale to assess frailty accurately. Previous studies regarding valve intervention have used it showing an ability to predict clinical outcomes [11,12,13,14,15]. However, its applicability in the usual clinical practice is still limited. Three conditions are necessary for a proper assessment: enough time, a physical space (clinical consult) and a professional trained explicitly for this purpose [23]. Therefore, considering the SPPB scale’s strengths and controlling its limitations, we designated the FRESAS study protocol with a specific consult. The same person trained explicitly in geriatric evaluation had enough time to evaluate all participants. 

The prevalence of frailty in our sample of patients was 19.6%. This result is somewhat lower than the ones reported in other registries with the same age group. This difference is probably explained by the different frailty definitions used in the other studies [9,10,11,12,13,14,24,25,26]. Goudzward et al. used the “Erasmus Frailty Score” (MMSE, grip strength, nutrition, BADL and IADL), obtaining a 28% prevalence of frailty in elderly patients undergoing TAVR [13]. Skaar et al. [14] evaluated frailty in TAVR patients with the “GA Frailty Score” (MMSE, nutrition, BADL, IADL, weight loss, standing ability, comorbidity assessed by the Charlson index and affective state) and a prevalence of 24% of frailty was found. A combined application of the different scales (Fried, Fried +, SPPB, Bern, Columbia and EFT) showed prevalence ranging from 12% (using the Rockwood scale) after TAVR to 74% (using SPPB scales) after SAVR [15]. In short, the use of different scales and definitions to evaluate frailty results in significant variability among various studies. Differences in the baseline characteristics and the prevalence of frailty between the FRESAS study and the most relevant studies are shown in Table 4. 

The FRESAS study identified the Barthel index, age and presence of AF as predictors of frailty. AF and age seem to act as risk factors and markers of frailty in our population and may share similar biological mechanisms. Molecular changes mediated by substances, such as cytokines and interleukins, have already been associated with AF and aging, increasing the risk of frailty [27,28]. Furthermore, our study shows a statistically significant association between frailty and functional dependency evaluated with the Barthel index. It has been proposed that frailty is a state that precedes disability among a “functional continuum” (robust-prefrail-frail-dependent) as aging occurs [29]. This hypothesis could explain the statistical association found. 

Patients with severe symptomatic aortic stenosis present limited physical activity attributed to the valvulopathy’s clinical symptoms (e.g., dyspnea, dizziness, asthenia). Moreover, there are no specific programs to encourage exercise in this subgroup of patients [30,31]. Accurate detection of frailty may provide an opportunity to safely intervene in elderly patients by implementing “prehabilitation programs” directed to reverse this geriatric syndrome and transform them into robust subjects. Our future line of work involves prehabilitation programs to turn frail patients into robust and improve outcomes.

Our study has certain limitations. It is an observational study in a single center. Only one person assessed frailty, with the inherent limitation to extrapolate this protocol to other centers. Moreover, frailty is a complex geriatric syndrome and not all of its components can be evaluated with one scale. In this regard, SPPB does not include malnutrition which is a strong marker of frailty. We believe that determining the actual prevalence of frailty in these patients in routine clinical practice, with the proposed systematic protocol in the FRESAS study, would help future patient assessment and guide clinical/interventional decisions. However, further studies are granted, as no definite frailty cut-offs have yet been established. Another limitation is that we have yet to develop prehabilitation programs to not only correctly identify frail patients but, to make changes in their functional status.

## 5. Conclusions

The prevalence of frailty in cohort or patients ≥75 years old with severe symptomatic AS undergoing transcatheter or surgical AVR using our standardized protocol was 19.6%. The evaluation by specifically trained personnel using a validated scale in an outpatient-based consult may reduce the variability in estimating the prevalence of frailty in elder patients with severe symptomatic AS. 

Functional status (evaluated by the Barthel index), age and the presence of AF are independent predictors for the development of frailty in patients ≥75 years with severe symptomatic AS.

## Figures and Tables

**Table 1 jcm-10-02345-t001:** Baseline characteristics of the patients included in the study.

	Total (*n* = 286)	Non-Frail (*n* = 230)	Frail (*n* = 56)	*p* Value
Demographic				
Age (years)	84 ± 4	3.7 ± 3.9	85.3 ± 4.31	0.007
Sex (Female), *n* (%).	175 (61.2)	136 (59.1)	39 (69.6)	0.148
Clinical				
Hypertension, *n* (%).	221 (77.3)	174 (75.7)	47 (83.9)	0.185
Dyslipidemia, *n* (%).	156 (54.5)	131 (57.0)	25 (44.6)	0.097
Atrial fibrillation, *n* (%).	95 (33.2)	65 (28.3)	30 (53.6)	0.000
Heart failure, *n* (%).	87 (30.4)	58 (25.2)	29 (51.8)	0.000
Pulmonary hypertension, *n* (%).	83 (29.2)	65 (28.5)	18 (32.1)	0.592
Diabetes mellitus, *n* (%).	71 (24.8)	57 (24.8)	14 (25.0)	0.973
Depression, *n* (%).	66 (23.1)	53 (23.0)	13 (23.2)	0.978
COPD, *n* (%).	62 (21.7)	48 (20.9)	14 (25.0)	0.501
Thyroid disease, *n* (%).	55 (19.2)	40 (17.4)	15 (26.8)	0.11
Chronic kidney disease, *n* (%).	52 (18.2)	42 (18.3)	10 (17.9)	0.944
AMI/ACS, *n* (%).	50 (17.5)	40 (17.4)	10 (17.9)	0.934
Cardiac surgery, *n* (%).	44 (15.4)	39 (17.0)	5 (8,9)	0.133
Stroke, *n* (%).	38 (13.3)	29 (12.6)	9 (16.1)	0.494
Peripheral vascular disease, *n*(%).	25 (8.8)	22 (9.6)	3 (5.4)	0.314
MCI/dementia, *n* (%).	12 (4.2)	7 (3.1)	5 (8.9)	0.049
	1.4 ± 1.3	1.4 ± 1.2	1.8 ± 1.5	0.011
Cardiological Parameters				
EuroSCORE-II.	3.81 ± 3.14	3.71 ± 2.96	4.20 ± 3.79	0.308
Preserved LVEF, *n* (%).	224 (81.8)	185 (84.5)	39 (70.9)	0.020
Aortic valve area (cm^2^).	0.71 ± 0.17	0.71 ± 0.17	0.69 ± 0.16	0.724
Mean gradient (mmHg).	44.93 ± 14.35	45.10 ± 14.64	44.19 ± 13.16	0.693
Peak gradient (mmHg).	72.81 ± 21.97	72.77 ± 22.17	72.95 ± 21.31	0.963
Laboratory parameters				
Hemoglobin (g/dL).	13.2 ± 1.9	12.9 ± 1.9	12.2 ± 1.7	0.014
Albumin (g/L).	44 ± 8.6	44.6 ± 9.2	41.3 ± 4.7	0.011
eGFR (mL/min/1.73 m^2^).	57 ± 18	57.6 ±17.6	54.3 ± 19.7	0.217
Creatinine (mg/dL).	1.2 ± 0.7	1.1 ± 0.7	1.2 ± 0.8	0.601
Folic acid (µg/dL).	7 ± 4.9	7 ± 4.2	7 ± 4.9	0.936
B12 vitamin (pg/mL).	503.2 ± 305.7	503.2 ± 284	512 ± 381.3	0.83
TSH (mU/L).	2.3 ± 1.8	2.2 ± 1.8	2.3 ± 1.9	0.697
25OH-D3 (ng/mL).	17.1 ± 10.8	17.3 ± 10.7	16.4 ± 11.1	0.638
Functional, nutritional and mental variables				
MNA.	11.6 ± 1.8	11.8 ± 1.6	10.8 ± 2.2	0.002
Barthel index.	92.9 ± 11.5	95.6 ± 7.1	81.8 ± 17.9	0.000
Lawton index in men.	4.1 ± 1.3	4.3 ± 1.1	2.9 ± 1.6	0.003
Lawton index in women.	6.3 ± 2.2	6.8 ± 1.7	4.3 ± 2.4	0.000
MMSE.	26.7 ± 3.4	27.2 ± 2.6	24.5 ± 5.1	0.000
Yesavage geriatric depression.	3.2 ± 2.8	3 ± 2.8	3.9 ± 2.9	0.025

COPD: Chronic obstructive pulmonary disease, ACS/AMI: acute coronary syndrome/acute myocardial infarction, MCI: mild cognitive impairment, NYHA: New York Heart Association, LVEF: left ventricle ejection fraction, eGFR: estimated glomerular filtration rate, TSH: thyroid-stimulating hormone, 25OH-D3: Vitamin D, MNA: Mini Nutritional Assessment Short-Form; SPPB: Short Performance Physical Battery, MMSE: Mini Mental State Examination.

**Table 2 jcm-10-02345-t002:** Multivariable logistic regression.

	OR	95% Confidence Interval	*p* Value
Age	1.128	1.017–1.252	0.023
Atrial fibrillation	2.326	1.047–5.165	0.038
Heart Failure	1.258	0.505–3.310	0.622
Anemia	1.416	0.580–3.456	0.444
Cognitive impairment	0.529	0.094–2.979	0.470
Left ventricle ejection fraction	1.771	0.655–4.788	0.260
Charlson Index	1.171	0.837–1.640	0.356
Hemoglobin	1.011	0.796–1.286	0.926
Albumin	0.908	0.815–1.011	0.079
Mini-nutritional assessment	1.113	0.873–1.419	0.389
Yesavage depression scale	1.017	0.868–1.193	0.834
Lawton	0.932	0.759–1.145	0.505
Minimental state examination	0.966	0.853–1.094	0.585
Barthel index	0.916	0.872–0.962	0.000

**Table 3 jcm-10-02345-t003:** Multivariable linear regression.

	β	*p* Value
Constant	5.95	0.025
Age	−0.08	0.005
Atrial fibrillation	−0.582	0.016
Charlson Index	−0.228	0.010
Yesavage depression scale	−0.089	0.036
Barthel index	0.108	0.001

**Table 4 jcm-10-02345-t004:** Baseline characteristics and prevalence of frailty among different studies.

	Fresas	Afilalo [15]	Ungar [21]	Rodriguez-Pascual [9]	Pegaso [19]	Green [10]	Arnold [11]	Huded [12]	Bureau [20]	Goudzward [13]	Skaar [14]
*n*	286	1020	71	606	928	244	2830	191	116	213	142
Female (%)	61.2	41	62	57.9	58.8	48.4	45.5	49	49.1	53.5	54
Age	83.9	82	85.4	82.9	84.2	86.2	83.3	82.4	86.2	83	83.4
Arterial hypertension (%)	77.3	-	83.1	80	76.6	88.9	-	81	-	82.1	-
Dyslipidemia (%)	54.5	-	-	54.1	42.1	-	-	63	-	67.8	-
Diabetes (%)	24.8	28	26.8	30.9	26.4	29.1	36.6	35	-	34.1	-
Acute coronary syndrome (%)	17.5	22	23.9	21.3	12.8	-	-	-	-	19.7	24
Heart failure (%)	30.4	-	-	36.7	50.5	99.2	-	67	-	-	-
Atrial fibrillation (%)	33.2	33	-	35.8	30.6	-	43.8	41	-	-	32
Anemia (%)	26.9	-	-	-	-	-	-	-	-	-	-
Chronic kidney disease (%)	18.2	-	7	22	-	13.9	-	28	-	46.5	4
Chronic obstructive pulmonary disease (%)	21.7	17	15.5	21.5	15.9	42.2	-	14	-	23.9	22
Mini-nutritional assessment - malnourishment (%)	11.6	-	-	-	-	-	-	-	-	-	-
	2.8	-	-	-	-	-	-	-	4.3	11.7	-
- Barthel- Limitation of the basic activities of daily living (%)	92.9	-	-	-	-	-	-	-	-	-	-
	27.6	25	-	28.8	50.6	29.5	16.7	-	10.4	31.5	-
- Lawton - Limitation of the instrumental activities of daily living (%)	5.8	-	-	-	-	-	-	-	-	-	-
	52.3	47	-	56.9	-	-	-	-	58.6	43.2	-
Short performance physical battery	8.4	7	5.7	-	-	-	-	-	-	-	-
Frailty (%)	19.6	12–74	-	49	-	45.1	59.8	33	-	28.6	24
- Minimental state examination - Cognitive impairment (%)	26.6	-	-	26.6	-	-	-	-	-	-	26.3
	14.1	18	-	-	-	-	-	-	17.2	34.7	44
- Yesavage - Depression (%)	3.2		-	4.6	-	-	-	-	-	-	-
	27.1	32	-	-	-	-	-	-	-	-	-

The highest value for each variable is shown among the different series.

## Data Availability

Data is available upon request.

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
