# Peer review of "Frailty Assessment in a Cohort of Elderly Patients with Severe Symptomatic Aortic Stenosis: Insights from the FRailty Evaluation in Severe Aortic Stenosis (FRESAS) Registry"

_jcm, 2021, doi:10.3390/jcm10112345_

Round 1
Reviewer 1 Report
Dear authors,
We are pleased to review the above-mentioned manuscript. The ambition of evaluating the prevalence of frailty in your patient groups by setting up an observational, longitudinal and prospective study is interesting.
Nevertheless, there are some aspects we would recommend to overthink:
-Has the study led to any consequences/recommendations?
A clarification of intentions which led to the project of analysing the prevalence of frailty in your population as well as the display, if existent, of certain consequences concerning the treatment or further contact with frail patients as a result of assessing the patient´s frailty would be advantageous.
Or has the main aim been to exclusively depict factors which can predict frailty (like the age, the Barthel Index and atrial fibrillation, p.1 ll 22+23)?
You reveal that patients characterised as frail are at higher risk of suffering delirium, falls and deconditioning (p.1, ll 35-36) after AVR and emphasize that the frailty-evaluation is „ essential to select an adequate treatment and improving outcomes in elderly patients with severe AS “(p. 1, ll 36 + 37).
But did you treat frail patients differently in order to prevent these adverse events (so that the diagnosed frailty justified any action recommendations)? Maybe, it was possible to observe these adverse events in your study, as well? The clinical need of your study design has to be adressed more clearly.
Or did the assessment of frailty help the heart team in choosing the appropriate intervention modality, that, for instance, frail patients were more likely to be treated non surgically? (p.1, l 32) or that an avoidance of „therapy that could be futile “(p. 2, l 59) resulted?
-Your frailty-definition:
Throughout reading the manuscript there are multiple questions arising regarding to your perception of „frailty “/ your concept and understanding of frail patients itself.
The fact that you establish your own “FRESAS study protocol”, as a “specific protocol” in order to be able to measure the prevalence of frailty in your patient cohort (p. 6, ll 200-204 : “available to estimate the actual prevalence in these patients “; p. 9 ll 290-293 : “ the prevalence of frailty (…) using our standardized protocol was 19,6 %”) leads to confusion when actually using the SPPB to declare frailty (p. 2, ll 67+68 :“according to SPPB evaluation, we divided the patients into two groups: Group A (frail, with an SPPB score of 6 or less) and Group B ( not frail, SPPB >6)”) and to diagnose the number of 56 frail patients in 286 study members what leads to the study´s stated frailty -prevalence of 19,6 % (p. 1 ll 23-24).
It is understandable that you critique the “lack of uniformity” (page 1, l 39; page 2 ll 45 +46) in literature when talking of frailty. Also, it is, then, comprehensible that the actual stated prevalence of frailty in the different studies might vary because of this missing exact definition and strict criteria to diagnose frailty in patients (p.2, ll 68-69; page 6 ll 214-216). But in this context, it seems not coherent, that you, however, describe an „internationally accepted “(p.3, l 117) questionnaire called „Short performance physical battery “(SPPB) to categorize your patients in either the frail or the non-frail group, to quantify the prevalence in your specific study and to compare this to other studies.
The change of the SPPB-Scale by forming two categories out of three (by minimizing the before existent categories „robust, prefrail and frail “into „frail and not frail “, p.3, ll 121+122) does not reinforce the impression that any comparability and uniformity to other studies is purposed. The criticism of the lack of a uniformity, again, is totally acceptable but should be overthought when, after all, naming a “widely recognized” (p.3, l 113) tool for assessing frailty in the following.
Furthermore, the presentation of the frailty score in both of your groups in more detail would have been relevant (especially when you put importance on the “evaluation of the degree of frailty” page 1, ll 14+15 and ll 30+31).
You mention that the mean SPPB score was 8,45 (in terms of the original SPPB scale categorized as “prefrail” with a SPPB score 7-9) but as a reader you might question what the distribution of frailty in each group looks like.
Maybe the frail patients are classified as “frail” by achieving a score nearby six and might therefore almost be categorized as “non- frail”. This thought evokes curiosity because it leads to further considerations of maybe possible existent transient patient factors, which, by treatment or elimination, could result in a patient achieving a non-frail condition.
What we think of in this context, for example, is the fact the anaemia is significantly more likely to occur in the frail group. This, by considering that equilibrium is one of the three items of the SPPB scale (p. 3, l 120) appears to be logical (when recapitulating pathophysiology).
But does this not open great possibilities for interventions in order to prevent adverse events occurring more often in frail patients and to, for example, “reduce the length of stay, readmissions, functional deterioration and death (p. 1, ll36+37).
-Background of your exclusion criteria:
The reader may be interested in finding out about the reasons which have led to the exclusion of patients younger than 75 years from your study (page 1, ll 64-65). What was the background of this certain cut off?
With regards to the exposed significant correlation of age and the presence of frailty (page 1, ll.22-23) it could have been important to involve younger patients to underline your statement by verifying a lower fraction of frailty when examining these younger patients or to investigate an enlarging percentage of frail patients while analysing patients from younger to older age.
-The protocol:
-the ambition of creating a specific protocol as presented in the manuscript and instructing a certain trained person, who constantly throughout the study is responsible for assessing the patient’s status (p.2 ll 80-85) concerning the listed criteria should be acknowledged.
Also, the intended uniformity and prevention of variability caused by different observers by guaranteeing the presence of the exact same person throughout the study should be honoured.
But could this also mean that subjectivity should be considered?
Furthermore, the belonging of the criteria in the listed groups may be rethought.
Could, for example, dyslipidaemia also or even more appropriate be allocated to the category laboratory.
Why do you choose preserved LVEF rather than the mean EF and the standard deviation? Why do focus on Albumin so obviously and do not name parameters like NT-proBNP?
And why do you list anaemia as well as haemoglobin?
As a reader you are definitely interested in the explanations and reasons that have led to this explicit selection of categories in your protocol.
Lastly, we would like to recommend to fulfil corrections in the following sections:
-Table 1. Baseline characteristics of patients included in the study.
- “clinical”
-Please correct the percentage of Heart Failure listed in the non-frail group.
- “laboratory parameters”
-Please look at the correctness of the measurement units.
Final comment: True novelty of the presented study seems hard to tell.
Reviewer 2 Report
This is an interesting study on frailty scores
main concern is methodologic
- the two group analysed differer significantly, not only for the variables reported but also for other main potential underlying factors which have not been listed or are simply hidden. This requires a degree of strictness in the statistical analysis warranting at least normalization or even propensity matching
- There is concern that the multivariate model is overfitted considering the amount of variables inserted into the model to predict the primary outcome which is actually a binary outcome. Despite Holmers coefficient is good, the model could be underpowered. This is a recurrent issue in cardiac surgery studies. Please see
http://dx.doi.org/10.1016/j.jtcvs.2016.07.046
- no insights on the significance of Barthel index is provided. This is once again a combined score which includes many variables and they should be taken into account in the statistical analysis as requiring many observations/events
- the claim on the specific consultation/allocated time and space is interesting but this advantage in comparison to other frailty indexes should be demonstrated on the basis of the data
- the speculation on the fear of exercising in the discussion is out of scope. Authors should further explain to make the reader understanding the relation of the statement with the general context of the manuscript
Reviewer 3 Report
In this paper, Solla-Suárez et al. want to determine the true prevalence of frailty in severe aortic stenosis, based on a complete and standardized evaluation in a single center. A proper evaluation of frailty is mandatory to predict beneficial effects of aortic valve replacement. The efforts to standardize the evaluation of frailty with only 1 operator in this article should be acknowledged.
About the content:
- The announced aim of the study was to determine the true prevalence of frailty. However, I do not see how this objective was reached, as frailty was only determined with SPPB. I think this is the biggest flaw in this article.
First, SPPB may not be the best criterion to evaluate frailty, as it was shown by Afilalo et al in JACC (2017). EFT may be a better predictor of mortality and disability, which are the expected endpoints. Second, the true aim of the study seems to be the determination of factors linked to frailty evaluated with SPPB, but the results lack novelty (age, AF, Barthel index). The objective should be rephrased, and maybe redirected for more innovation. In my opinion, 1-year mortality is also essential.
- Methods: why were the patients who died excluded from the study? If they were evaluated before the intervention, shouldn’t they be included anyway?
- Methods: SPPB can classify patients into frail/prefrail/robust, but what made the authors choose a binary classification frail/robust?
- Discussion: the idea developed l254-260 that patients should benefit prehabilitation programs based on frailty assessment could be interesting, but quite risky in symptomatic aortic stenosis. Instead, maybe you could expose the idea of intensive rehabilitation programs and specific nutrition after surgery/TAVR?
About the form:
- The introduction should be improved. I understand that a lack of uniformization exists in the evaluation of frailty, but this idea is redundant all throughout the introduction (l33 + l39 + l45). The ideas exposed in the introduction could be more synthetized.
- Methods: l 113-118 would rather correspond to the introduction
- Numbers should be rounded, such as age (84 instead of 83.99), aortic valve area (1 digit) and gradients (max 1 digit), laboratory tests (1 digit) and the functional nutritional mental variables (1 digit).
- Results: multivariable analysis results should be revised to be clearer. Line 172 belongs to methods.
- Discussion: there are 32 references in the text, but only 31 in the bibliography?
- An asterisk could be added in the tables for significant variables, to improve readability
- Many English mistakes or mistranslation were made, I would recommend English-native proofreading, and the use of simpler sentences. Some of the mistakes I noticed:
- L37: improve instead of improving
- L64: “those” should be removed
- L74: remove “one”
- L76: do you mean “lost to follow-up”?
- L154: please do not start the sentence with a number (170)
- L204: able instead of available
- L239: use instead of employment
- L251: remove a “could”
I acknowledge the amount of work that was necessary to obtain the data, but the objectives and the results are not up to this work. Therefore, major changes are required.
Round 2
Reviewer 1 Report
Dear authors,
thank you for your additional effort. The mansucript has improved. I still miss substantial novelty. Like I stated above - language is another problem, e.g. “In this context and given the possibility that exercise in patients with severe AS may develop symptoms like angina, syncope, or sudden death. “(page 7, ll 268-272) does not make sense.
Kind regards
Author Response
Reviewer # 1
Dear reviewer,
We thank you for the comments you have provided that have helped to make substantial changes and we believe the quality of the work has improved in this context. The document was sent for editing and several changes were made regarding style, wording and English in general. These changes have been tracked and highlighted in yellow. We feel that the general idea of our work is more clearly delivered thanks to these modifications. Moreover, we answer your final comments in the following lines.
Author’s response # 1
We thank the reviewer for this observation and we have removed lines 268-272 accordingly.
Reviewer 3 Report
I would like to thank Solla-Suárez et al. for their substantial work to improve the manuscript. I am satisfied that the main idea went from “actual proportion of frailty” to “standardized evaluation of frailty”.
However, I still have difficulties to understand the novelty of this paper: if it is the evaluation of frailty in AS, it was already done previously on a larger scale. The “standardization”, with a single nurse evaluating all the patients, also has its own drawbacks, as it may lack objectivity and does not evaluate reproducibility. It may also not be applicable in every center, considering the time required to assess a patient.
The use of SPPB as a gold standard of frailty is also debatable, and therefore the factors influencing SPPB may not be the factors indeed influencing frailty (whatever the definition). As there is no mortality evaluation, the modifications in treatment are only subjective, and are not expressed in mortality benefits.
Therefore, these points should be addressed or added in the limitations.
Lines 127-130 still do not belong to the methods, they should be (re)moved.
Author Response
Author’s response #1
We agree with the reviewer that because the present work is an observational study in a single center and since one person performed the evaluation, it carries certain limitations. We have made changes to our limitation section according to your suggestions:
“Our study has certain limitations. It is an observational study in a single center. Only one person assessed frailty, with the inherent limitation to extrapolate this protocol to other centers. Moreover, Frailty is a complex geriatric syndrome and not all of its components can be evaluated with one scale. In this regard, SPPB does not include malnutrition which is a strong marker of frailty.”
Reviewer’s comment #2
Lines 127-130 still do not belong to the methods, they should be (re)moved.
Author’s response # 2
We thank the reviewer for this observation and we have removed lines 127-130 accordingly.